# The role of sociodemographic and psychosocial variables in early childhood development: A secondary data analysis of the 2014 and 2019 Multiple Indicator Cluster Surveys in the Dominican Republic

**Laura V. Sánchez-Vincitore**[1], **Arachu Castro**[2]*

**1** Neurocognition and Psychophysiology Laboratory, Universidad Iberoamericana (UNIBE), Santo Domingo, Dominican Republic, **2** Department of International Health and Sustainable Development, Tulane University School of Public Health and Tropical Medicine, New Orleans, Louisiana, United States of America

* acastro1@tulane.edu

## Abstract

The association between sociodemographic factors—poverty, lack of maternal schooling, being male at birth—, childhood developmental delay, and poor educational outcomes has been established in the Dominican Republic (DR). However, family moderating factors present or introduced to buffer sociodemographic factors effects on early childhood development (ECD) are still unknown. We conducted a secondary analysis of the DR's 2014 and 2019 Multiple Indicator Cluster Surveys. We had four study aims: 1) confirm the relationship between socioeconomic position (SP), parenting practices, and ECD; 2) determine if a sociodemographic model predicted ECD; 3) determine if a psychosocial model (family child-rearing practices, discipline, and early childhood stimulation) predicted ECD above and beyond the sociodemographic model; 4) explore mothers' beliefs about physical punishment and its relationship with ECD and psychosocial variables. We found that both models predicted ECD significantly, but the psychosocial model explained more variance than the sociodemographic model (6.3% in 2014 and 4.4% in 2019). The most relevant sociodemographic predictors were SP (explaining 21.6% of ECD variance in 2014 and 18.6% in 2019) and mother's education (explaining 13.9% in 2014 and 14.1% in 2019). The most salient ECD psychosocial predictors were: negative discipline, number of children's books at home, stimulating activities at home, and attendance to an early childhood education program. The predicting weights of the independent variables were similar for both years. These results have multiple implications for social programs that aim to improve children's potential in contexts of poverty. Although the results show a protective effect of psychosocial factors, sustainable and large-scale interventions should not be limited to just buffering effects, but to solve the underlying problem, which is that poverty prevents children from reaching their developmental potential and exposes them to life-long greater risk for chronic disease. Addressing delays early in life can therefore contribute to achieving health equity.

**Data Availability Statement:** The raw data are available at http://mics.unicef.org/surveys after registration at http://mics.unicef.org/ by entering a username and password and requesting access to conduct secondary data analysis of childhood development indicators of the Dominican Republic National Household Survey with Multiple Purposes —Multiple Indicator Cluster Survey (MICS) 2014 and 2019. Any researcher is able to access these data in the same manner as the authors, who did not have any special access privileges that others would not have.

**Funding:** The study was funded by the 2018 Carol Lavin Bernick Faculty Grant Program at Tulane University made to AC, who was also funded through gifts from the Zemurray Foundation for her position as the Samuel Z. Stone Chair of Public Health in Latin America at the Tulane School of Public Health and Tropical Medicine. The funders had no role in the design of the study, collection, analysis, and interpretation of data nor in writing the manuscript.

**Competing interests:** The authors have declared that no competing interests exist.

## Introduction

Early childhood development (ECD) is a multisectoral concern that involves global health, social assistance, education, and other sectors, particularly in contexts of poverty [1–4]. Approximately 249 million children in lower- and middle-income countries might not develop fully due to stunting and living in moderate or extreme poverty [5]. There is a 10% prevalence of cognitive delay in preschoolers in lower and middle-income countries, which calls for evidence-based prevention strategies to avoid the loss of developmental potential [6].

The Dominican Republic is no exception, facing multiple challenges regarding childhood development and other educational outcomes, partly due to adverse early childhood experiences. According to the most recent multidimensional childhood poverty index, 49.3% of Dominican children lived in poverty and 7.7% lived in extreme poverty in 2012 [7]. Neonatal and under-five mortality rates were high, with 19 and 29 deaths per 1,000 live births, respectively, in 2018 [8]. Moreover, 21.5% of newborns were born to an adolescent mother in 2019, the highest percentage in Latin America and the Caribbean [9]. Additionally, sociocultural childrearing practices sometimes promote psychological aggression and physical punishment to discipline children, evidenced by 2019 household survey data stating that 64% of Dominican children aged 1 to 14 years have experienced such a method of discipline [10].

### Factors that negatively impact childhood development

Numerous studies have demonstrated that negative experiences in childhood may alter children's development and brain chemistry [3, 11–19], and these effects are detectable starting in early childhood [20]. Impairments to brain development carry implications for cognitive, social, and behavioral deficits that may affect children as they enter adolescence and adulthood [3]. Two of the most studied factors are exposure to stress and poverty.

Children's stress responses to these adverse exposures have been characterized as positive, tolerable, or toxic, depending on their intensity and duration and their potential to impact physiologic regulation [21]. Mild to moderate adverse exposures can trigger brief and positive stress responses that can promote growth, particularly in the presence of a supportive caregiver [3]. In the absence of protection from supportive caregivers, the exposure to more adverse circumstances can lead to toxic stress—that is, excessive or prolonged activation of biological stress responses that can alter brain structure and function, and lead to impaired linguistic, cognitive, and social-emotional skills, as well as a life-long greater risk for chronic disease [3, 22]. Therefore, delay during early childhood is a structural driver of inequities in the conditions of daily life and a source of health inequities along the social gradient [23].

Children who live in poverty are more likely to be exposed to forms of deprivation such as neglect, economic hardship, and undernutrition, to threats such as abuse, caregiver substance use, violence from or among caregivers, community violence, as well as to other forms of early adverse experience, than children from more affluent, supportive households [1–4]. Grantham-McGregor and colleagues described two pathways in which poverty affects childhood development and school achievement [24]. The first pathway, related to the lack of food, hygiene, and overall health in poor settings, can lead to stunting, which can cause developmental delay and poor school achievement. The second pathway, which represents the effects of poverty on caregivers, such as stress, depression, low responsivity, and low education, can lead to inadequate care—which can cause stunting—and limited stimulation at home—which affects both childhood development and school achievement [24]. The correlation between socioeconomic position (SP) and childhood development exists throughout the world, including high-, middle-, low- [25, 26], and very low-income countries [25]. Experiencing poverty

during childhood can also impact cognitive development [3, 27–31]. This triggers a cascade of negative effects in developing other complex educational skills, such as language, literacy, and math [32].

## Factors that positively impact childhood development

Multiple factors can buffer the pervasive effects of poverty on ECD, as development is sensitive to the context in which children live [33]. A nurturing environment during childhood can improve brain function, protect against disease, and lead to good health during adulthood [20, 34, 35]. A study conducted in 28 developing countries found a positive correlation between positive caregiving and the human development index (HDI) [36]. Frongillo and colleagues found that family care behavior moderated the effects of SP on children's literacy and numeracy skills [37]. On the other hand, negative caregiving practices in the form of physical punishment have a detrimental effect in childhood development [38] by increasing the risk of childhood anxiety [39] and behavioral problems [40]. Regarding the availability of stimulating objects at home, a study conducted in 42 countries found that the number of books at home (family scholarly culture) correlates positively with academic achievement, even when controlling for parents' education, occupation, and family SP [41]. The study showed stronger effects at the lower end of the family scholarly culture variable; that is, the effect of adding extra books on academic achievement was larger when there were fewer books at home. In addition, literacy and numeracy are associated with the home literacy environment and the availability of stimulating activities at home [37].

Providing a nurturing environment can be challenging in settings of poverty and high social inequality [35]. Governments, international cooperation agencies, non-governmental organizations, and others offer intervention programs for children at risk to overcome this difficulty. The most effective interventions are those that provide service directly to children and support parents with their childrearing practices by offering both information and building parenting skills [42, 43]. Home visit programs have shown to be effective in improving the quality of the home environment, in increasing the number of playing activities with children at home [44], and in attaining moderate to large gains in social-emotional and language development [45].

## Study aims

Although the association between poverty and developmental delay in early childhood has been established in the Dominican Republic [7], the role of other factors that may impact childhood development and may be present in families of any socioeconomic position is unknown. Therefore, this study's goal was to understand the extent to which the availability of positive environment and childhood stimulation buffers the effects of poverty on childhood development among Dominican children.

Our study's first aim was to confirm the well-studied relationship between SP, parenting practices such as discipline and stimulation at home, and childhood development. The second aim was to determine if a sociodemographic model that accounts for demographic factors that are well known to affect childhood development—such as wealth, mother's education, sex, and age—can predict early childhood development. The third aim was to determine if a psychosocial model that includes social factors that result from the social interaction between children, their caregivers, and the environment can predict ECD above and beyond the sociodemographic model. The fourth aim explored mothers' beliefs of physical punishment and its relationship with childhood development and psychosocial variables.

## Materials and methods

We conducted a secondary analysis of the 2014 Multiple Indicator Cluster Survey Round 5 (MICS5) [46] and the 2019 MICS Round 6 (MICS6) [10] databases for the Dominican Republic. MICS is a household survey administered by local governments from low- and middle-income countries assisted by UNICEF. For MICS 2014, a total of 33,097 household surveys were completed. From these, 20,187 completed the childhood questionnaire. A total of 8,039 children were between 36–59 months of age, and finally, 7,356 cases selected the same child for both early childhood development and discipline information. For MICS 2019, a total of 34,982 household survey were completed. From these, 8,503 completed the childhood questionnaire. A total of 3,399 cases were selected for both early childhood development and discipline information since they were in the selected age range (36–59 months old).

The cases consisted of children between 36 to 59 months of age (50% were female). To obtain sociodemographic and psychosocial information from the participants, the following inclusion criteria had to be met: 1) caregivers had completed the MICS Under Five Children's questionnaire modules (child's information panel module and early childhood development module); 2) caregivers had completed the MICS Household questionnaire modules (selection of one child for child discipline module and household characteristics module); and 3) the child for which there were data on ECD on the Under Five Children's questionnaire matched the selected child for the discipline module of the Household questionnaire. The variables used are included in Table 1.

For the first aim, we conducted Pearson's correlation analyses to evaluate the relationship between family socioeconomic position and ECD and to explore the relationship between family SP and the other variables assessed in this study. To explore if the magnitude of the correlations changed from 2014 to 2019, we conducted a Fisher's $r$ to $z$ transformation to determine statistical significance with observed $z$ statistic. The analyses accounted for household survey weights and significance threshold was established at $p < .05$.

For the second and the third aims, we conducted a two-step hierarchical multiple regression analysis to predict early childhood development. Fig 1 presents a visual representation of the model. Significance threshold was established at $p < .05$. We entered the predictors in two blocks using the Enter process. The first block (second aim) consisted of sociodemographic factors (child's age in months, child's sex at birth, family SP, and mother's education level). The second block (third aim) consisted of both sociodemographic factors and psychosocial factors (stimulating activities at home, stimulating objects at home (homemade and store-bought), number of books at home, attendance to childhood program, positive discipline, and negative discipline). To visually compare the sociodemographic and psychosocial models that predict ECD from both years, we plotted the standardized regression coefficients of each model.

For the fourth aim, we conducted a series of analyses of covariance (ANCOVA) comparing childhood development and psychosocial scores between mothers who agree with physical punishment vs. mothers who do not agree with physical punishment, controlling for SP. All analyses were conducted on SPSS V.20. Given the large sample size, and only a few outliers (ranging from 0% to 0.26% across the variables), compensation for skewness was not necessary.

## Patient and public involvement

There was no patient or public involvement in this study.

**Table 1. MICS variables used in the analysis, missing values, and internal consistency, Dominican Republic 2014 and 2019.**

| Household questionnaire | | Missing values and internal consistency |
|---|---|---|
| Child's age | This variable is expressed in months. | |
| Child's sex | The scoring of this variable was 0 for females, 1 for males. | |
| Family socioeconomic position | This variable categorizes families into five wealth quintiles from 1 (poorest) to 5 (richest). | |
| Mother's education level | This variable assigns an ordinal value to maternal education: | Missing data: |
| | 2014: 1 = none; 2 = primary; 3 = secondary; 4 = higher education | 2014 = 0.07% |
| | 2019: 0 = none; 1 = primary; 2 = secondary; 3 = tertiary | 2019 = 0.32% |
| **Childhood development module** (for children under five years of age) | | |
| Early childhood development index | This is a series of ten yes/no statements regarding the child's observed behaviors administered to the mother or primary caregiver. Observed behaviors obtained a score of 1 and not observed behaviors obtained a score of 0. The behaviors included: (a) literacy and numeracy statements (the child identifies at least ten letters of the alphabet; the child reads at least four simple, popular words; the child knows the name and recognizes the symbol of all numbers from 1–10); (b) physical development (the child picks up small objects with two fingers; the child is sometimes too sick to play); (c) approaches to learning (the child follows simple directions, the child does something independently); (d) and social and emotional development (the child gets along well with other children; the child kicks, bites or hits other children or adults; the child gets distracted easily). The answers for negative statements (the child is sometimes too sick to play; the child kicks, bites of hits other children or adults; and the child gets distracted easily) were reversed to keep the scoring valence consistent. The childhood development index was computed by averaging all childhood development scores for each answer. The index ranged from 0 to 1. Missing data were excluded from calculating the index. | Missing data: <br><br> 2014 = 0% <br><br> 2019 = 0.03% <br><br> ECD index had poor internal consistency ($\alpha$ = .5). |
| Availability of stimulating activities at home | This instrument is a series of yes/no statements regarding developmentally appropriate activities that parents or other people do with the child. This includes reading, storytelling, singing, taking walks, playing, counting, and naming. The informer reported if the mother, father, and/or another person does each activity with the child scoring 0 for not doing the activity and 1 for doing the activity. The stimulating activities at home variable was computed by averaging the stimulating activities scores for each answer, assuming that the more activities a child does, the higher the score; and the more people doing the activities at home, the higher the score. The scores for the variable stimulating activities at home ranged from 0 to 1. | No missing data. |
| Availability of stimulating homemade toys and objects at home | This is a yes/no questionnaire regarding the child having access to homemade toys or stimulating objects at home. The informer reported whether the child had access to any of the two types of stimulating objects, scoring 0 for not having access, and 1 for having access. The availability of stimulating home-made objects at home variable was computed by averaging the stimulating object scores for homemade toys and stimulating objects at home. The scores ranged from 0 to 1. | Missing data: <br><br> 2014: 0.39% <br><br> 2019: 0% |
| Availability of stimulating store-bought toys | This is a yes/no question regarding the child having access to store-bought toys. The availability of stimulating store-bought toys was computed with the single item that measured access to store-bought toys. The scores ranged from 0 to 1. | Missing data: <br><br> 2014 = 0.08% <br><br> 2019 = 0% |
| Number of books at home | This variable represents the amount of children's books at home. This is a numeric variable. | Missing data: <br><br> 2014 = 0.07% <br><br> 2019 = 0% |
| Attendance to early childhood education center | The variable was scored 0 for not attending and 1 for attending. This includes private or government facilities, kindergarten, or daycare. | Missing data: <br><br> 2014 = 0.62% <br><br> 2019 = 0.06% |
| **Child discipline module**: The module consists of eleven statements regarding methods of child discipline and is an adapted version from the Parent-Child Conflict Tactics Scale [47]. The adult participant was asked to indicate whether they have used any of the discipline methods during the past month. For this study, we categorized the methods into two categories: positive discipline and negative discipline. | | |

(*Continued*)

**Table 1.** (Continued)

| Household questionnaire | | Missing values and internal consistency |
|---|---|---|
| *Positive discipline* | Averaging answers to the following statements computed this variable: took away privileges, explained why the behavior was wrong, and gave the child something else to do. The positive discipline variable scores ranged from 0 to 1. | No missing data. |
| | | The instrument had acceptable levels of internal consistency ($\alpha = .6$). |
| *Negative discipline* | Averaging answers to the following statements computed this variable: shook the child; shouted, yelled or screamed at the child; spanked, hit or slapped child on the bottom with bare hands; hit the child on the bottom or elsewhere with a belt, brush, stick, etc.; called child dumb, lazy or another name; hit or slapped the child on the face, head or ears; hit or slapped the child on the hand, arm or leg; and beat child up as hard as one could. The negative discipline variable ranged from 0 to 1. | Missing data |
| | | 2014 = 0% |
| | | 2019 = 0.06% |
| | | The instrument had acceptable levels of internal consistency ($\alpha = .6$). |
| *Beliefs about physical punishment* | The childhood discipline module asks mothers if they consider physical punishment as an appropriate strategy to correctly discipline a child. The variable was coded as 0 if mother does not agree with physical punishment, and 1 if mother agrees with physical punishment. N = 7,288 (2014) and N = 2,127 (2019). | |

**Source**: Based on the Multiple Indicator Cluster Survey for the Dominican Republic [10, 46].

## Ethics statement

This study used de-identified secondary data that are publicly available and that cannot be re-identified. Therefore, it poses no violation of confidentiality. After registration in the MICS

**Fig 1. Hierarchical regression models to predict childhood development.**

database, the authors obtained permission from UNICEF to download and use the database for research purposes. The MICS survey responds to local regulations and protocols regarding data collection and fieldwork. The Dominican Republic does not demand to obtain approval from an ethical review board. However, the MICS protocol requires all data to be kept strictly confidential, including secure storage of records and databases with no identifiers, and requires the interviewer to obtain appropriate informed consent from survey respondents. The protocol states that information is strictly confidential by the Dominican Law 5096 [48].

## Results

After obtaining the descriptive statistics from the MICS 2014 and 2019 samples, shown in Table 2, we conducted a t-test to compare the ECD scores results between the samples. The results show that the 2019 sample obtained a higher development score than the 2014 (2014 mean = .67; 2019 mean = .68); however, the effect size was negligible ($t(10,752) = 3.21$, $p = .003$, $d = .06$).

For the first aim, we conducted a Pearson's correlation analysis to evaluate the relationship between ECD and family socioeconomic position, accounting for household weights. As shown in Table 3, we found a significant positive correlation between the two variables in both samples, indicating that the higher the family SP, the higher the childhood development score.

**Table 2. Descriptive statistics for tested variables, MICS Dominican Republic 2014 and 2019.**

| 2014 | N | Min | Max | M | SD |
|---|---|---|---|---|---|
| D. Early childhood development index | 7,356 | 0 | 1 | 0.67 | 0.15 |
| SD1. Child's age in months | 7,356 | 36 | 59 | 48.15 | 6.84 |
| SD2. Child's sex (% male) | 7,356 | 0 | 1 | 0.51 | 0.50 |
| SD3. Family socioeconomic position | 7,356 | 1 | 5 | 2.58 | 1.39 |
| SD4. Mother's level of education | 7,351 | 1 | 4 | 2.82 | 0.87 |
| PS1. Stimulating activities at home | 7,356 | 0 | 0.75 | 0.19 | 0.13 |
| PS2a. Stimulating homemade toys at home | 7,327 | 0 | 0.67 | 0.36 | 0.25 |
| PS2b. Stimulating store-bought toys at home | 7,350 | 0 | 1 | 0.92 | 0.27 |
| PS3. Number of books at home | 7,351 | 0 | 10 | 1.11 | 1.86 |
| PS4. Attendance to early childhood program (%) | 7,310 | 0 | 1 | 0.36 | 0.48 |
| PS5. Positive discipline | 7,356 | 0 | 1 | 0.45 | 0.35 |
| PS6. Negative discipline | 7,356 | 0 | 1 | 0.17 | 0.18 |
| **2019** | **N** | **Min** | **Max** | **M** | **SD** |
| D. Early childhood development index | 3,398 | 0 | 1 | 0.68 | 0.15 |
| SD1. Child's age in months | 3,399 | 36 | 59 | 47.54 | 6.92 |
| SD2. Child's sex (% male) | 3,399 | 0 | 1 | 0.50 | 0.50 |
| SD3. Family socioeconomic position | 3,399 | 1 | 5 | 2.54 | 1.38 |
| SD4. Mother's level of education | 3,388 | 0 | 3 | 2.00 | 0.86 |
| PS1. Stimulating activities at home | 3,399 | 0 | 0.75 | 0.22 | 0.14 |
| PS2a. Stimulating homemade toys at home | 3,399 | 0 | 0.67 | 0.40 | 0.24 |
| PS2b. Stimulating store-bought toys at home | 3,399 | 0 | 1 | 0.94 | 0.23 |
| PS3. Number of books at home | 3,399 | 0 | 10 | 0.96 | 1.76 |
| PS4. Attendance to early childhood program (%) | 3,397 | 0 | 1 | 0.48 | 0.50 |
| PS5. Positive discipline | 3,397 | 0 | 1 | 0.53 | 0.34 |
| PS6. Negative discipline | 3,397 | 0 | 1 | 0.20 | 0.19 |

Notes: N = sample size; Min = Minimum; Max = Maximum; M = Mean; SD = Standard Deviation.

**Source**: Based on data from the Multiple Indicator Cluster Survey for the Dominican Republic [10, 46].

**Table 3. Comparison of correlation magnitudes between family socioeconomic position and other variables in 2014 vs. 2019.**

|  | SD3. Family socioeconomic position | | z | p |
|---|---|---|---|---|
|  | **2014** | **2019** |  |  |
| D. Early childhood development index | 0.29*** | 0.23*** | 3.1 | 0.001 |
| PS1. Stimulating activities at home | 0.3*** | 0.28*** | 1.03 | 0.303 |
| PS2a. Stimulating homemade toys and objects | -0.12*** | -0.09*** | -1.72 | 0.085 |
| PS2b. Stimulating store-bought toys | 0.25*** | 0.2*** | 2.48 | 0.013 |
| PS3. Number of books at home | 0.42*** | 0.37*** | 3.07 | 0.002 |
| PS4. Attendance to early childhood program | 0.39*** | 0.21*** | 9.48 | < .001 |
| PS5. Positive discipline | 0.12*** | 0.11*** | 0.48 | 0.631 |
| PS6. Negative discipline | -0.02 | -0.13*** | 4.93 | < .001 |

Notes:

* p < .05;

** p < .01;

*** p < .001.

z = z test score; p = statistical significance. Weighted by sample weight.

**Source**: Based on data from the Multiple Indicator Cluster Survey for the Dominican Republic [10, 46].

The magnitudes of these correlations were statistically different between the two samples, indicating that there was a smaller correlation in 2019 than in 2014 between ECD and socioeconomic position ($z = 3.10$, $p = .001$). To understand the extent to which variations of SP co-occur with variations of other intervening factors in childhood development, we conducted a Pearson's correlation analysis to explore the relationship between SP and 1) early childhood stimulation, and 2) child discipline. As shown in Table 3, we found significant positive correlations between SP and early childhood stimulation both in 2014 and 2019: the availability of stimulating activities at home; numbers of books at home; availability of store-bought toys; and attendance to an early childhood education center. That is, children from more affluent homes have access to more stimulating resources and activities than children from less affluent homes. We found a significant negative correlation between socioeconomic position and the availability of stimulating homemade toys and objects at home, although the magnitude of the correlation was small. Both in 2014 and 2019, we found a positive significant correlation between SP and the use of positive discipline by parents and caregivers, also shown in Table 3. In 2014, we did not find a significant correlation between SP and the use of negative discipline, whereas in 2019 we found a negative correlation between socioeconomic position and the use of negative discipline.

For the second and third aims, before conducting the hierarchical multiple regression analysis, we checked the linearity, normality, homoscedasticity, and multicollinearity assumptions for this particular test in MICS 2014 and MICS 2019. The assumptions of linearity, normality, and homoscedasticity were met. Variance inflation factor statistics scored below 10 for all the variables, confirming no multicollinearity. Homoscedasticity assumption was tested through the scatterplot of standardized residuals. The hierarchical multiple regression analysis, step one corresponded to a sociodemographic model. We entered the independent variables: child's age in months (SD1), child's sex at birth (SD2), family socioeconomic position (SD3), and mother's level of education (SD4). Step two corresponded to a sociodemographic + psychosocial model to which we added the psychosocial variables: stimulating activities at home (PS1), stimulating homemade toys and objects at home (PS2a), stimulating store-bought toys at home (PS2b), the number of books at home (PS3), attendance to an early childhood

education program (PS4), positive discipline at home (PS5), and negative discipline at home (PS6). Regression statistics are reported in Table 4.

The hierarchical multiple regression analysis showed that, at step one, the sociodemographic model contributed significantly to the regression model ($F(4,7271) = 296.59$, $p < .001$, in 2014; $F(4, 3175) = 111.78$, $p < .001$, in 2019) and accounted for 14.0% of the variance in ECD in 2014 and 12.4% in 2019. At step two, the sociodemographic + psychosocial model contributed significantly to the regression model ($F(11, 7271) = 168.67$, in 2014; $F(11,3175) = 57.92$, $p < .001$, in 2019), explaining an additional 6.3% of the variance in 2014 and 4.4% in 2019. This change in $R^2$ was significant ($F(7, 7260) = 82.23$, $p < .001$, in 2014; F(7, 3174) = 23.91, p < .001, in 2019). Together, the sociodemographic and psychosocial variables accounted for 20.4% of the variance in ECD in 2014 and 16.8% in 2019.

Most regression coefficients were significant in both models (except for stimulating homemade toys at home and positive discipline in 2014; and stimulating store-bought toys and positive discipline in 2019). Most of the coefficients had positive valences, which means that they had positive effects on childhood development. However, some independent variables had negative regression coefficients: male sex at birth and negative discipline (both in 2014 and 2019), and the availability of homemade toys in 2019. For the sociodemographic model, being born male explained 4.8% of childhood development variance in 2014, and 6.2% in 2019. For the sociodemographic + psychosocial model, being born male explained 3.6% of childhood development variance in 2014 and 4.3% in 2019. Negative discipline negatively impacted child development, explaining 12.6% of the variance of ECD in the sociodemographic + psychosocial model in 2014 and 13.3% in 2019. Having homemade toys and objects at home negatively impacted childhood development, explaining 5.5% of the variance of ECD in the psychosocial model. This result was only found in 2019, as this predictor was not significant in the 2014 model.

The most relevant sociodemographic predictors of ECD were socioeconomic position and mother's education: SP explained 21.6% of the ECD variance in 2014 and 18.6% in 2019; mother's education explained 13.9% of ECD variance in 2014 and 14.19% in 2019. The most relevant psychosocial predictors in 2014 were: 1) attendance to an early childhood education program, which uniquely explained 15.5% of the variance; 2) negative discipline, which uniquely explained 12.6% of the variance; 3) number of children's books at home, which uniquely explained 12.2% of the variance; 4) availability of store-bought toys at home, which uniquely explained 8.0% of the variance. In 2019, they were: 1) negative discipline, which uniquely explained 13.3% of the variance; 2) the number of children's books at home, which uniquely explained 10.6% of the variance; 3) stimulating activities at home, which uniquely explained 7.4% of the variance; 4) attendance to an early childhood education program, which uniquely explained 5.9% of the variance in ECD.

Fig 2 shows a visual representation that compares the standardized coefficients from both models. The variance explained by the sociodemographic + psychosocial model includes the predictive power of the factors that already belonged to the sociodemographic model. This is more evident for three factors: child's age in months, family socioeconomic position, and mother's level of education.

For the fourth aim, we explored the implications of the survey question that asked mothers or primary caregivers to indicate whether they identified with the following statement: "Children must be physically punished." We divided our sample into two groups: caregivers who believed that physical punishment is necessary and those who believed that physical punishment is not necessary. We conducted a series of analyses of covariance (ANCOVA) to test differences between the groups on ECD, early childhood stimulation, and child discipline, controlling for socioeconomic position. We included SP as a covariate given its correlation

**Table 4. Summary of hierarchical regression analysis for variables predicting early childhood development, MICS Dominican Republic 2014 and 2019.**

| 2014 (N = 7,272) | b | SE b | β | CI Lower | CI Higher | $R^2$ | $\Delta R^2$ |
|---|---|---|---|---|---|---|---|
| Step 1: Sociodemographic model | | | | | | 0.140 | 0.140 |
| Constant | 0.34 | 0.01 | | | | | |
| SD1. Child's age in months | 0.00 | 0.00 | 0.198*** | 0.00 | 0.01 | | |
| SD2. Child's sex at birth (male) | -0.02 | 0.00 | -0.048*** | -0.02 | -0.01 | | |
| SD3. Family socioeconomic position | 0.02 | 0.00 | 0.216*** | 0.02 | 0.03 | | |
| SD4. Mother's level of education | 0.03 | 0.00 | 0.139*** | 0.02 | 0.03 | | |
| Step 2: Sociodemographic + psychosocial model | | | | | | 0.204 | 0.063 |
| Constant | 0.39 | 0.01 | | | | | |
| SD1. Child's age in months | 0.00 | 0.00 | 0.153*** | 0.00 | 0.00 | | |
| SD2. Child's sex at birth (male) | -0.01 | 0.00 | -0.036** | -0.02 | -0.01 | | |
| SD3. Family socioeconomic position | 0.01 | 0.00 | 0.106*** | 0.01 | 0.01 | | |
| SD4. Mother's level of education | 0.01 | 0.00 | 0.051*** | 0.01 | 0.01 | | |
| PS1. Stimulating activities at home | 0.06 | 0.01 | 0.047*** | 0.03 | 0.09 | | |
| PS2a. Stimulating homemade toys and objects | 0.00 | 0.01 | 0.006 | -0.01 | 0.02 | | |
| PS2b. Stimulating store-bought toys | 0.05 | 0.01 | 0.080*** | 0.04 | 0.07 | | |
| PS3. Number of books at home | 0.01 | 0.00 | 0.122*** | 0.01 | 0.01 | | |
| PS4. Attendance to early childhood program | 0.05 | 0.00 | 0.155*** | 0.04 | 0.06 | | |
| PS5. Positive discipline | 0.01 | 0.01 | 0.020 | 0.00 | 0.02 | | |
| PS6. Negative discipline | -0.11 | 0.01 | -0.126*** | -0.13 | -0.09 | | |
| **2019 (N = 3,176)** | b | SE b | β | CI Lower | CI Higher | $R^2$ | $\Delta R^2$ |
| Step 1: Sociodemographic model | | | | | | 0.124 | 0.124 |
| Constant | 0.37 | 0.02 | | | | | |
| SD1. Child's age in months | 0.01 | 0.00 | 0.208*** | 0.00 | 0.01 | | |
| SD2. Child's sex at birth (male) | -0.02 | 0.01 | -0.062*** | -0.03 | -0.01 | | |
| SD3. Family socioeconomic position | 0.02 | 0.00 | 0.186*** | 0.02 | 0.03 | | |
| SD4. Mother's level of education | 0.03 | 0.00 | 0.141*** | 0.02 | 0.03 | | |
| Step 2: So***ciodemographic + psychosocial model | | | | | | 0.168 | 0.044 |
| Constant | 0.40 | 0.02 | | | | | |
| SD1. Child's age in months | 0.00 | 0.00 | 0.188*** | 0.00 | 0.01 | | |
| SD2. Child's sex at birth (male) | -0.01 | 0.01 | -0.043** | -0.02 | 0.00 | | |
| SD3. Family socioeconomic position | 0.01 | 0.00 | 0.106*** | 0.01 | 0.02 | | |
| SD4. Mother's level of education | 0.02 | 0.00 | 0.088*** | 0.01 | 0.02 | | |
| PS1. Stimulating activities at home | 0.08 | 0.02 | 0.074*** | 0.04 | 0.12 | | |
| PS2a. Stimulating homemade toys and objects | -0.04 | 0.01 | -0.055** | -0.06 | -0.02 | | |
| PS2b. Stimulating store-bought toys | 0.03 | 0.01 | 0.03 | 0.00 | 0.05 | | |
| PS3. Number of books at home | 0.01 | 0.00 | 0.106*** | 0.01 | 0.01 | | |
| PS4. Attendance to early childhood program | 0.02 | 0.01 | 0.059** | 0.01 | 0.03 | | |
| PS5. Positive discipline | 0.00 | 0.01 | 0.00 | -0.01 | 0.02 | | |
| PS6. Negative discipline | -0.11 | 0.01 | -0.133*** | -0.14 | -0.08 | | |

Notes:

* p < .05;

** p < .01;

*** p < .001.

β = Standardized coefficients. Weighted by sample weight

**Source**: Based on data from the Multiple Indicator Cluster Survey for the Dominican Republic [10, 46].

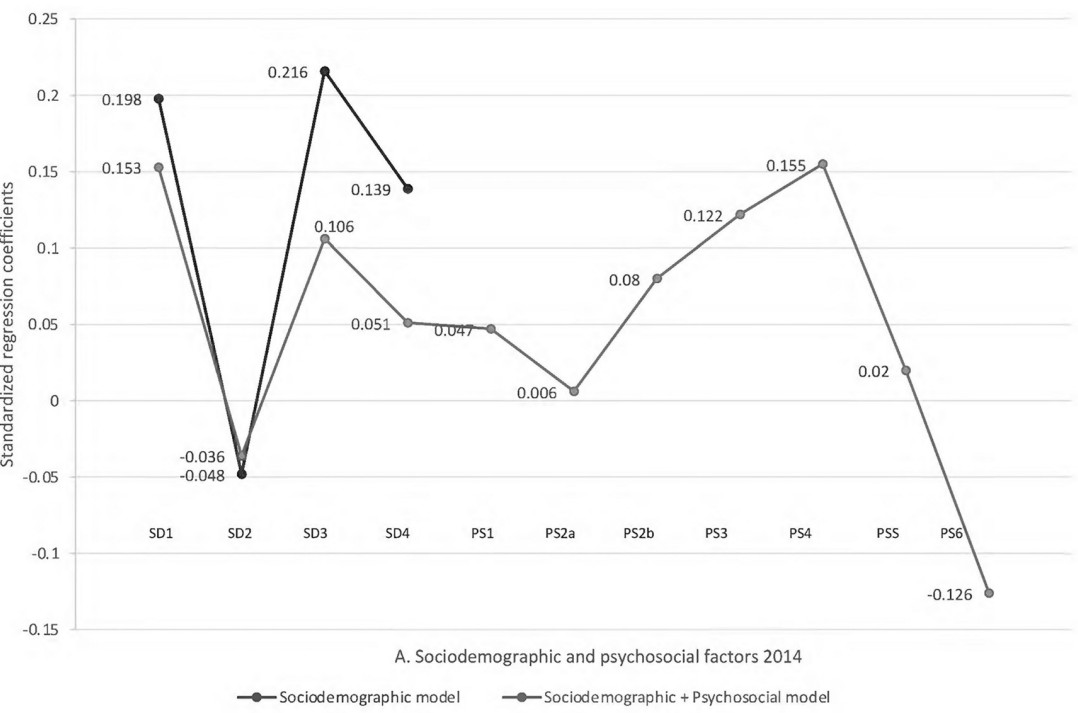

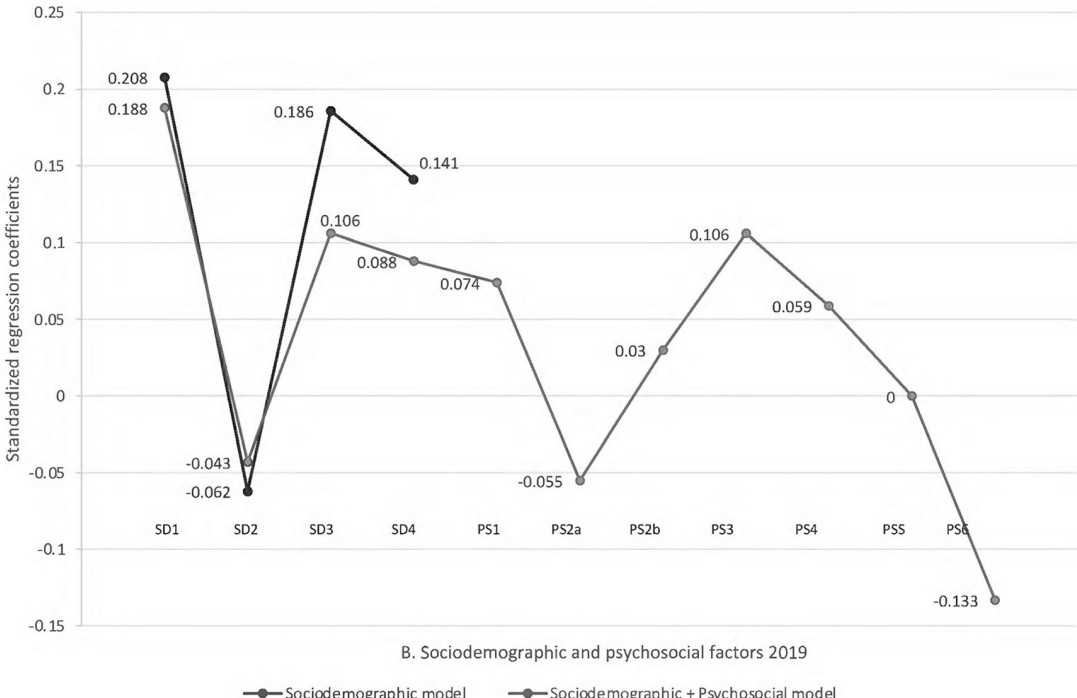

**Fig 2. Regression coefficients for models that predict early childhood development in the Dominican Republic, 2014 and 2019.** SD1 = Child's age in months; SD2 = Child's sex (male); SD3 = Family socioeconomic position; SD4 = Mother's level of education; PS1 = Stimulating activities at home; PS2a = Stimulating homemade toys and objects; PS2b = Stimulating store-bought toys; PS3 = Amount of books at home; PS4 = Attendance to early childhood program; PS5 = Positive discipline; PS6 = Negative discipline. Source: Based on data from the Multiple Indicator Cluster Survey for the Dominican Republic [10, 46].

**Table 5. Descriptive statistics by caregivers who agree and who do not agree that children must be physically punished.** MICS Dominican Republic 2014 and 2019.

| 2014 | Caregiver does not agree with physical punishment | | | Caregiver agrees with physical punishment | | | | | |
|---|---|---|---|---|---|---|---|---|---|
| | N | M | S.D. | N | M | S.D. | F | p | η² |
| D. Early childhood development index | 6,826 | 0.68 | 0.11 | 462 | 0.64 | 0.12 | 25.28 | 0.000 | 0.003 |
| PS1. Stimulating activities at home | 6,826 | 0.2 | 0.09 | 462 | 0.17 | 0.09 | 20.51 | 0.000 | 0.003 |
| PS2a. Stimulating homemade toys and objects | 6,801 | 0.34 | 0.18 | 458 | 0.4 | 0.18 | 24.14 | 0.000 | 0.003 |
| PS2b. Stimulating store-bought toys | 6,821 | 0.94 | 0.16 | 461 | 0.89 | 0.24 | 18.54 | 0.000 | 0.003 |
| PS3. Number of books at home | 6,822 | 1.24 | 1.4 | 461 | 0.94 | 1.31 | 3.66 | 0.060 | 0.001 |
| PS4. Attendance to early childhood program | 6,784 | 0.42 | 0.35 | 459 | 0.38 | 0.36 | 0.01 | 0.940 | 0.000 |
| PS5. Positive discipline | 6,826 | 0.45 | 0.25 | 462 | 0.56 | 0.26 | 57.82 | 0.000 | 0.008 |
| PS6. Negative discipline | 6,826 | 0.16 | 0.12 | 462 | 0.32 | 0.15 | 381.67 | 0.000 | 0.05 |
| 2019 | Caregiver does not agree with physical punishment | | | Caregiver agrees with physical punishment | | | | | |
| | N | M | S.D. | N | M | S.D. | F | p | η² |
| D. Early childhood development index | 1,875 | 0.69 | 0.15 | 181 | 0.64 | 0.17 | 14.62 | 0.000 | 0.007 |
| PS1. Stimulating activities at home | 1,876 | 0.23 | 0.14 | 181 | 0.19 | 0.14 | 11.71 | 0.001 | 0.006 |
| PS2a. Stimulating homemade toys and objects | 1,876 | 0.38 | 0.24 | 181 | 0.34 | 0.25 | 7.89 | 0.005 | 0.004 |
| PS2b. Stimulating store-bought toys | 1,876 | 0.96 | 0.18 | 181 | 0.93 | 0.26 | 3.08 | 0.079 | 0.001 |
| PS3. Number of books at home | 1,876 | 1.19 | 1.88 | 181 | 1.20 | 2.31 | 1.26 | 0.262 | 0.001 |
| PS4. Attendance to early childhood program | 1,875 | 0.53 | 0.50 | 180 | 0.49 | 0.53 | 0.40 | 0.526 | 0.000 |
| PS5. Positive discipline | 1,876 | 0.58 | 0.32 | 181 | 0.62 | 0.34 | 4.55 | 0.033 | 0.002 |
| PS6. Negative discipline | 1,876 | 0.20 | 0.17 | 181 | 0.31 | 0.20 | 79.11 | 0.000 | 0.037 |

Notes: M = mean; S.D. = standard deviation. F = ANOVA test's value; η² = effect size. Covariate—SD3 socioeconomic position. Weighted by sample weight.

**Source**: Based on data from the Multiple Indicator Cluster Survey for the Dominican Republic [10, 46].

with most tested variables and to prevent confounding results. Table 5 shows the descriptive statistics by group.

The results indicate that children whose caregivers do not agree with physical punishment compared with children whose caregivers agree with physical punishment (controlling for SP and considering sample weight) had: better development scores ($F(1,7287) = 25.28$, $p < .001$, in 2014; $F(1,2055) = 14.62$, $p < .001$, in 2019); more availability of stimulating activities at home ($F(1,7287) = 20.59$, $p < .001$, in 2014; $F(1,2056) = 11.71$, $p < .001$, in 2019); more availability of stimulating homemade toys and objects at home ($F(1, 2056) = 7.89$, $p = .005$, in 2019). In 2014, there were no significant differences between the groups on the availability of stimulating objects and number of books at home, when controlling for SP and considering sample weights; in 2019, there were no significant differences between the groups on the availability of store-bought toys and number of books at home. There were no significant differences in attendance to early childhood education program neither in 2014 nor 2019. Finally, caregivers who agree with physical punishment engage more both in positive discipline ($F(1, 7287) = 57.82$, $p < .001$, in 2014; $F(1, 2056) = 4.55$, $p < .033$, in 2019) and negative discipline ($F(1,7287) = 381.68$, $p < .001$, in 2014; $F(1, 2056) = 79.12$, $p < .001$, in 2019) compared with caregivers who do not agree with physical punishment, when controlling for socioeconomic position.

## Discussion and conclusion

This study's first aim confirmed that the well-established notion that wealth and inequality impact childhood development is valid in the Dominican Republic. More specifically, we

found that Dominican children from lower socioeconomic positions had lower childhood development scores than children from more affluent homes, both in 2014 and 2019. In the second and third aims, we found that a predictive model that includes both sociodemographic (age, sex at birth, socioeconomic position, mother's level of education) and psychosocial factors (childhood stimulation and child discipline) predict early childhood development above and beyond the sociodemographic factors alone, both in 2014 and 2019. Psychosocial factors such as attendance to childhood education programs, availability of children's books at home, and availability of stimulating activities at home were the strongest positive predictors of ECD. On the other hand, the use of negative discipline was the strongest negative predictor of childhood development. Therefore, both psychosocial factors and sociodemographic factors explain childhood development. In other words, engaging in childhood stimulation—especially the attendance at childhood education programs, availability of books at home, and activities—and not using negative discipline buffer the effects of socioeconomic position and maternal education on childhood development.

The correlation magnitude between family socioeconomic position and childhood development decreased from 2014 to 2019. This was also the case for correlations between family SP and 1) stimulating store-bought toys, 2) number of books at home, and 3) attendance to early childhood program. The correlation magnitude between family SP and negative discipline increased from 2014 to 2019. In fact, this correlation was not statistically significant in 2014, but it was in 2019. This means that most variables shared less variance with SP in 2019 than 2014, except for negative discipline, in which SP seems to gain relevance.

Similar results have been found elsewhere. In a study with data from 26 low- and middle-income countries, literacy and numeracy were associated with attendance at an early childhood program, home literacy, and stimulating activities at home, and the effect of socioeconomic position on development was moderated by family care behavior [37]. A study conducted by Tran et al. with data from 35 countries found that parental engagement in learning activities at home, hard punishment, and attendance to early childhood education moderate the effects of family poverty on development, but only in countries of low and medium human development index, and not in countries of high HDI [49]. The Dominican Republic is categorized as of high HDI since 2010, four years before the 2014 MICS survey, but of medium HDI when adjusted for inequality [50]. We hypothesize that this discrepancy could be due to the large role that inequality plays in preventing the poorest children from thriving in countries of high HDI but with a large share of the income concentrated within the wealthiest 10% of the population—35.4% in the Dominican Republic in 2010–17 [50].

This study's third aim found that both in 2014 and 2019 positive discipline was more likely to be used by parents and caregivers from higher socioeconomic positions. In 2014, negative discipline—one of the strongest factors that impact childhood development—was used regardless of socioeconomic position and, in 2019, negative discipline was more likely to be used by parents and caregivers from lower socioeconomic positions. Other studies have shown that negative discipline—specifically physical punishment—predicts young children's aggressive behavior [40], which could then create a vicious cycle of aggressive childhood behavior that triggers aggressive discipline methods in adulthood. Lansford and Deater-Deckard suggested that there are country-specific factors associated with the use of violence to discipline children, evidenced by a large variability of the reported use of negative discipline in a study of 24 low- and middle-income countries in MICS household surveys [51]. These sociocultural aspects should be studied in depth in the Dominican Republic, as almost two thirds of children from 1 to 14 years of age had experienced negative discipline during the month prior to the survey in 2014 (63%) and 2019 (64%) [10, 46].

Our study's fourth aim found that, both in 2014 and 2019, children whose caregivers believe in physical punishment had lower development scores, showing the detrimental effects of violence in childhood development. Another interesting result was that caregivers who do not believe in physical punishment provide more stimulating activities at home than caregivers who believe in physical punishment, but not necessarily more books, toys, or stimulating objects. The actual social interaction within stimulating activities might improve children's behavior, and not so much the available props with which to play. This finding supports Worku and colleagues' work that reported moderate to large gains in social-emotional and language development from a home visit intervention that targeted mother-child play interaction [45]. Finally, we found that caregivers who believed in physical punishment used more positive but also more negative discipline methods. Therefore, positive and negative discipline methods are not mutually exclusive, as has been shown earlier in Caribbean countries [52], and a caregiver of a child with challenging behaviors might use multiple discipline strategies.

This study has some limitations. One of them is a threat to internal validity due to instrumentation. First, the MICS early childhood development instrument has poor internal consistency. This could be the result of ECD being a multidimensional construct. In fact, the instrument classifies early childhood development into four categories: literacy and numeracy development, physical development, social and emotional development, and approaches to learning. Some of these categories are measured with only two items, making it unsuitable for internal consistency testing. For example, the physical development subtest is measured by asking caregivers: a) if the child is too sick to play, and b) if the child can pick up a small object with two fingers. These two items are unrelated, and fewer items on a scale usually give lower scores. Notwithstanding the internal consistency limitation, this study's tested models were strong enough to yield significant results, which means that the effects are seen even with a weak instrument.

Second, the instrument uses the same items to assess children ranging from 36 to 59 months of age. Childhood development is a continuum that changes quickly during the first months of life, and, therefore, a 59-month-old child will outperform a 36-month-old child on any development test. We overcame this limitation by adding the independent variable age in months to the sociodemographic model, to account for differences across age groups.

Third, there are country-specific characteristics with which this instrument might interfere. For example, the instrument item "can read simple common words" is part of the developmental assessment. Still, in the Dominican Republic, the official literacy training occurs in first grade, when children are six years old, which is older than the MICS5 age group. This issue, reported elsewhere [53], places most Dominican children at risk of underscoring when the reality is that they have not been trained to respond to such stimuli just yet. However, it is important to mention that this instrument belongs to a household survey, whose intention is not to provide a comprehensive assessment of ECD, but as a monitoring tool to evidence broad progress and allow cross-country comparisons.

Fourth, the Dominican Republic adopted in 2021, to use at a national scale, the Dominican Child Development Measurement System (*Sistema de Medición del Desarrollo Infantil Dominicano*, SIMEDID), based on a study conducted in 2017 [54] that was an adaptation of the Malawi Developmental Assessment Tool [55]. Nationwide, systematic data collection using SIMEDID is expected to begin in 2022 for the first time in the country.

Fifth, evidence from various settings has demonstrated that malnutrition, including micronutrient deficiencies, may prevent optimal development among children and adolescents [56–59]. However, we could not add nutrition to the models because MICS5 and MICS6 only report nutrition variables for children ages 0 to 23 months old. To our knowledge, no research has specifically examined this phenomenon in the Dominican Republic, where stunting has

decreased among children under 5 years from 19.4% in 1991 to 6.9% in 2013 but is distributed unequally between children from the poorest wealth quintile (11.3%) and those from the richest (3.9%) [60, 61]. Additionally, household survey data from 2014 show that more than half of children aged 6 to 23 months did not have a minimum acceptable diet in 2014 [46]—in terms of nutritional diversity or frequency of meals as defined by UNICEF [62]; no data were available in 2019.

Finally, the obtained model only explained 20.4% of childhood development's variance in 2014 and 16.8% in 2019. This study does not account for confounding variables that might interfere with the relationships between the studied variables, or other additional variables that explain childhood development.

Notwithstanding these limitations, this study is relevant because it provides a better understanding of some specific characteristics of how childrearing in the Dominican Republic is associated with ECD and what types of activities should be prioritized to prevent childhood developmental delay. In recent years, the Dominican Republic has made enormous efforts to promote ECD, such as the creation in 2015 of the National Plan for Early Childhood Comprehensive Protection and Care and the National Institute for Early Childhood Comprehensive Care (INAIPI, for its acronym in Spanish) [63], which offers home visit programs and early childhood education centers in the areas of highest social vulnerability at no cost. Although the coverage does not yet reach the entire target population, it is an indication that the country is making progress.

For multisectoral approaches to be effective, they need to consider and integrate the social, economic, political, and cultural contexts, the environment for caregivers, and the types of nurturing care (health, nutrition, security and safety, responsive caregiving, and early learning) [64]—in such ways that they promote interventions that mitigate the multiple risk factors for developmental delays and that promote nurturing caregiving, including socio-emotional and cognitive stimulation [65]. Although the results of this study show a protective effect of psychosocial factors, sustainable and large-scale interventions should not be limited to just buffering, but to solve the underlying problem, which is that poverty prevents children from reaching their developmental potential and exposes them to lifelong greater risk for chronic disease. Addressing delays early in life can therefore contribute to achieving health equity.

## Supporting information

**S1 File. Alternative language abstract: Abstract in Spanish.**
(DOCX)

## Acknowledgments

We thank Carlos Ruiz-Matuk for his support during data analysis and Virginia Savage and Liz Tracy for assisting with literature review.

## Author Contributions

**Conceptualization:** Laura V. Sánchez-Vincitore, Arachu Castro.

**Formal analysis:** Laura V. Sánchez-Vincitore, Arachu Castro.

**Funding acquisition:** Arachu Castro.

**Methodology:** Laura V. Sánchez-Vincitore, Arachu Castro.

Writing – **original draft:** Laura V. Sánchez-Vincitore, Arachu Castro.

Writing – **review & editing:** Laura V. Sánchez-Vincitore, Arachu Castro.

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
