## [Decision Letter · Decision Letter 0]

28 Feb 2022

PGPH-D-22-00017

The role of sociodemographic and psychosocial variables in early childhood development: A secondary data analysis of the 2014 Multiple Indicator Cluster Survey in the Dominican Republic

Dear Dr. Castro,

Thank you for submitting your manuscript to PLOS Global Public Health. After careful consideration, we feel that it has merit but does not fully meet PLOS Global Public Health’s publication criteria as it currently stands. Therefore, we invite you to submit a revised version of the manuscript that addresses the points raised during the review process.

We look forward to receiving your revised manuscript.

Kind regards,

Biplab Kumar Datta, Ph.D.

Academic Editor

Journal Requirements:

1. Please provide separate figure files in .tif or .eps format only.  Please ensure that all files are under our size limit of 20MB.  

For more information about how to convert your figure files please see our guidelines: Once you've converted your files to .tif or .eps, please also make sure that your figures meet our format requirements: https://journals.plos.org/climate/s/figures

2. We see that your study includes live participants, but you have not included an Ethics Statement. Please update your manuscript file to include an Ethics Statement subsection to your Materials and Methods section. It should include:

iii) (for human participants or donors) - A statement that formal consent was obtained (must state whether verbal/written) OR the reason consent was not obtained (e.g. anonymity)

3. We have noticed that you have uploaded supporting information but you have not included a list of legends.  Please add a full list of legends for all supporting information files (including figures, table and data files) after the references list. 

Additional Editor Comments (if provided):

Please revise the manuscript in line with the reviewers' comments and concerns. Additionally, it is not mentioned how the standard errors were treated in the analyses. The MICS provides survey weights, which are usually incorporated in empirical analysis. If the results account for the complex survey weights, please clearly mention that in the methods section. If not, then update the results using MICS survey weights. Otherwise give proper justification why survey weights were not used. In such case, make sure to state how robust standard errors were obtained (for example, through clustering at appropriate geographic level).

Reviewers' comments:

Reviewer's Responses to Questions

**Comments to the Author**

1. Does this manuscript meet PLOS Global Public Health’s publication criteria? Is the manuscript technically sound, and do the data support the conclusions? The manuscript must describe methodologically and ethically rigorous research with conclusions that are appropriately drawn based on the data presented.

Reviewer #1: Yes

Reviewer #2: Partly

2. Has the statistical analysis been performed appropriately and rigorously?

Reviewer #1: I don't know

Reviewer #2: No

3. Have the authors made all data underlying the findings in their manuscript fully available (please refer to the Data Availability Statement at the start of the manuscript PDF file)?

Reviewer #1: Yes

Reviewer #2: Yes

4. Is the manuscript presented in an intelligible fashion and written in standard English?

Reviewer #1: Yes

Reviewer #2: Yes

5. Review Comments to the Author

Reviewer #1: 1. The literature review is hard to read since there are several papers with mixed findings. The authors should re-organize the literature review sorting the evidence and including sub-sections.

2. My main concern is on confounding and omitted variables. What is the reliability of the results? Certainly, the methods used in the paper cannot account for it. Even in the limitation section, the authors do not discuss it.

3. What are “few outliers”? The authors should be more precise.

4. The authors argue that “the assumptions of linearity, normality, and homoscedasticity were met, as evidenced by the residual and scatter plots”. Each of these assumptions can be proved using a specific statistical test rather than scatter plots.

Reviewer #2: Thank you very much for enabling me to review this paper. This is an interesting and large study which has looked at risk factors for poor development in children between 36 to 59 months of age and provides useful information about sociodemographic and psychosocial factors linked to poor development in Dominican Republic.

I have detailed some comments about the sections and I hope that these recommendations help as you progress this manuscript.

Introduction

The introduction is too long. I wonder if it might be shortened somewhat and more to the point where it justifies the gap in the literature and the reasons for the study. For example, the paragraph starting in line 117 is about interventions, such as home visits programs, and this is not the focus of the study.

Some specific points:

L53 – I think ECD is not only a global health concern, but a multisectoral concern, involving education, social assistance and other sectors.

L55 – Please check if the estimates of 249 million children not achieving fully development is based in all these factors (poverty, inequality, violence, stress, deficient care, lack od education and opportunities, among others) instead of stunting prevalence and the percentage of children living in extreme or moderate poverty.

I also suggest reviewing the study objectives, considering some comments below.

Methods

Sampling procedures and the data collection instrument used are well described.

However, it would be important to improve the description of the modeling steps, perhaps including a figure presenting the hierarchical model. The modeling process needs to be better described (a backward stepwise process?). Only the variables that reached statistical significance (p < .05) were maintained in the model when the psychosocial variables were introduced in the model?

Results

Some descriptions of the modeling steps were presented in the Results (lines 216-223; lines 229-232) but should be presented in Methods.

Some procedures presented in the results were not described in the methods: evaluation of the internal consistency of the ECD index, positive and negative discipline; the Pearson’s correlation analysis to explore the relationship between socioeconomic position and early childhood stimulation and the correlation between socioeconomic position and child discipline. The analysis according to the survey question "Children must be physically punished” was not described as a part of the research question. It should be included in the objectives and methods.

Discussion

The hierarchical multiple regression analysis showed that, at step one, the sociodemographic model contributed significantly to the regression model and accounted for 14.3% of the variance in ECD. When introducing the psychosocial variables, the new model explained an additional 5.3% of the variance. Together, the sociodemographic and psychosocial variables accounted for 19.6% of the variance in ECD. I’m not sure this means that psychosocial factors moderate the relationship between sociodemographic factors and childhood development or that engaging in childhood stimulation buffer the effects of socioeconomic position and maternal education on childhood development. It seems that both sociodemographic and psychosocial variables are associated with child development and together, explain better the outcome. To test the moderation effect, it would be interesting introduce interaction terms in the model.

6. PLOS authors have the option to publish the peer review history of their article (what does this mean?). If published, this will include your full peer review and any attached files.

**Do you want your identity to be public for this peer review?** For information about this choice, including consent withdrawal, please see our Privacy Policy.

Reviewer #1: No

Reviewer #2: No

---

## [Editor Report · Decision Letter 1]

9 May 2022

PGPH-D-22-00017R1

The role of sociodemographic and psychosocial variables in early childhood development: A secondary data analysis of the 2014 and 2019 Multiple Indicator Cluster Surveys in the Dominican Republic

Dear Dr. Castro,

Thank you for submitting your manuscript to PLOS Global Public Health. After careful consideration, we feel that it has merit but does not fully meet PLOS Global Public Health’s publication criteria as it currently stands. Therefore, we invite you to submit a revised version of the manuscript that addresses the points raised during the review process.

We look forward to receiving your revised manuscript.

Kind regards,

Biplab Kumar Datta, Ph.D.

Academic Editor

Journal Requirements:

Additional Editor Comments (if provided):

Thank you for addressing the reviewer comments. There some minor issues that need to be addressed before I can recommend this paper for publication. Those are as follows:

1. In response to reviewer-1’s comment, the VIF statistics were provided. The VIF takes care of the multicollinearity issue, not heteroskedasticity. Please respond to the reviewer’s comment on how the homoskedasticity assumption was met.

2. Please correct the typo in pg.3 lines 66-67. Please consider dividing the long sentence in lines 64-69 into two or more short sentences.

3. In the Background section – 4 aims were mentioned. In the methods section, it’s hard to follow how these aims were addressed. Please clearly outline what statistical analyses were performed to address each of the 4 aims. For example, for aim-1 following statistical analyses were performed ….. ; for aim-2 following statistical analyses were performed …

4. Include a sentence stating whether the correlation between ECD and family’s socioeconomic position in 2014 (i.e., 0.29) was different from that (i.e., 0.23) in 2019.

5. The Pearson’s correlation results presented in pg.12-13 are very difficult to follow. Please present the results in a table and describe the key findings in the text. In the table also indicate whether the correlation coefficients in 2014 were different from those in 2019. In the methods section state what tests were conducted to determine the differences in the correlation coefficients across two survey periods. Please indicate if the correlation analyses account for MICS survey weights. If not, provide justifications.

6. The sentence “Descriptive statistics are displayed in Table 2.” In pg.13 line 240 is redundant. It was already mentioned in pg.11 lines 212-213. Please drop one.

7. There are some redundancies in mentioning of the Pearson’s correlation test and presentation of results in pg.11 lines 209-212 and pg.12-13 lines 223-228. Please rearrange the text to effectively communicate your findings.

8. Remove the methodology of two-step hierarchical regression from the results section (pg.14 lines 249-257) in the methods section under respective aim.

9. What is beta in Table 3? Clearly state in the table foot note.

10. In the methods section clearly state how Figure 2 was created.

11. In the footnote of table 4 clearly state what are F and (eta)2.

12. Compare whether the development scores in 2014 were statistically different from those in 2019.

13. In the discussion section, discuss the results by the 4 specific aims. It will help the readers to understand the important contribution of your work.

14. Also please discuss how the estimates of 2014 were different (or not different) from that of 2019.

I will look forward receiving the final version of your manuscript.
---

## [Editor Report · Decision Letter 2]

29 Jun 2022

The role of sociodemographic and psychosocial variables in early childhood development: A secondary data analysis of the 2014 and 2019 Multiple Indicator Cluster Surveys in the Dominican Republic

PGPH-D-22-00017R2

Dear Dr. Castro,

We are pleased to inform you that your manuscript 'The role of sociodemographic and psychosocial variables in early childhood development: A secondary data analysis of the 2014 and 2019 Multiple Indicator Cluster Surveys in the Dominican Republic' has been provisionally accepted for publication in PLOS Global Public Health.

Best regards,

Biplab Datta, Ph.D.

Academic Editor